# Achieving Ultra-High Performance Concrete by Using Packing Models in Combination with Nanoadditives

**DOI:** 10.3390/nano11061414

**Published:** 2021-05-27

**Authors:** Jesús Díaz, Jaime C. Gálvez, Marcos G. Alberti, Alejandro Enfedaque

**Affiliations:** 1Departamento de Ingeniería Civil: Construcción, E.T.S de Ingenieros de Caminos, Canales y Puertos, Universidad Politécnica de Madrid, 28040 Madrid, Spain; jesus.diaz@upm.es (J.D.); marcos.garcia@upm.es (M.G.A.); alejandro.enfedaque@upm.es (A.E.); 2Lantania S.A, C/Sobrado, 2, 28050 Madrid, Spain

**Keywords:** particle packing, packing models, packing density, concrete, addition, nanoaddition, ultra-high-performance concrete

## Abstract

This paper describes the packing models that are fundamental for the design of ultra-high-performance concrete (UHPC) and their evolution. They are divided into two large groups: continuous and discrete models. The latter are those that provide the best method for achieving an adequate simulation of the packing of the particles up to nanometric size. This includes the interaction among the particles by means of loosening and wall coefficients, allowing a simulation of the virtual and real compactness of such particles. In addition, a relationship between virtual and real compactness is obtained through the compaction index, which may simulate the energy of compaction so that the particles are placed in the mold. The use of last-generation additives allows such models to be implemented with water–cement (*w*/*c*) ratios close to 0.18. However, the premise of maximum packing as a basic pillar for the production of UHPC should not be the only one. The cement hydration process affected by nanoadditives and the ensuing effectiveness of the properties in both fresh and hardened states according to the respective percentages in the mixture should also be studied. The characterization tests of the aggregates and additions (dry and wet compactness, granulometry, density and absorption) have been carried out in order to implement them numerically in the polydisperse packing model to obtain the compactness of the mixture. Establishing fixed percentages of nanoadditives in the calculation of the mixture’s compactness. The adequate ratio and proportion of these additions can lead to better results even at lower levels of compactness. The compressive strength values obtained at seven days are directly proportional to the calculated compactness. However, at the age of 28 days, better results were obtained in mixes with lower cement contents, fewer additions and lower compactness. Thus, mixes with lower cement contents and additions (silica fume and limestone filler) with a compactness of *φ* = 0.775 reached 80.1 MPa of strength at 7 days, which is lower than mixes with higher cement contents and number of additions (SF, limestone filler and nanosilica), which achieved a compactness of *φ* = 0.789 and 93.7 MPa for compressive strength. However, at 28 days the result was reversed with compressive strengths of 124.6 and 121.7 MPa, respectively.

## 1. Introduction

In the design of ultra-high-performance concrete (UHPC), the so-called “minimum defect” is established as the main objective. Such an objective entails creating a material with the minimum number of voids (microcracks and interconnected pores) in order to reach the potential strength of the components and increase the durability of the concrete. For this reason, an optimization of the packing of particles is established in the design by means of models that consider the individual compactness of each component and the maximum compactness achieved by the set of components. However, this maximum compactness, which is translated into mixing percentages, may not always be the most optimal in terms of hydration of paste and strength. Therefore, this study seeks to balance the maximum compactness achieved by means of packing models with the optimum ratio among the materials and by considering hydration processes. Additions, such as nanosilica or metakaolin, favor the formation of calcium silicate hydrate (C-S-H), which allows an improvement in strength and durability by reducing the porous network, though the percentages should be limited to certain values in order to achieve a greater degree of effectiveness [1,2,3,4]. Thus, if the content of additions exceeds specific percentages, the result may be a reduction in mechanical strength [3].

According to the ACI Committee 239 [5] and also the Federal Highway Administration (FHWA) [6], UHPC is a concrete with a compressive strength of at least 150 MPa. This administration includes eight performance characteristics: freeze–thaw durability, scaling resistance, abrasion resistance, chloride penetration, compressive strength, modulus of elasticity, shrinkage, and creep. When specific tensile strength and ductility properties are sought in the design, concrete is often referred to as an Ultra-High-Performance Fiber-Reinforced Concrete (UHPFRC). 

For example, in France, the NF P18-710 standard [7] defines UHPFRC concrete as: a concrete with high compressive strength and high post-cracking tensile strength, giving it a ductile behavior in tension, whose lack of brittleness makes it possible to design and produce structures and structural members without using reinforced steel. The required mechanical properties are as follows: characteristic compression strength *f_ck_* between 150 and 250 MPa, characteristic tensile strength *f_ctk_,_el_* greater than 6.0 MPa and sufficiently ductile behavior under tension in order to satisfy the following inequality:(1)$$\frac{1}{w_{0.3}}\cdot \int_{0}^{w_{0.3}}\frac{\sigma \left(w\right)}{1.25}dw\geq \max\left(0.4\cdot f_{ctm,el};3\;\mathrm{MPa}\right)$$
where $f_{ctm,el}$ is the mean value of the tensile limit of elasticity, *w*_0.3_ = 0.3 mm and *σ*(*w*) is the characteristic post-cracking stress as a function of the crack width *w*. In addition, these concretes are required to comply with properties related to durability: water porosity, coefficient of apparent diffusion of chloride ions and apparent gas permeability; additionally, they have a density of 2300–2800 kg/m^3^. On the other hand, when the main properties sought are related to durability in certain service states, the concrete is referred to as an “Ultra High Durability Concrete” (UHDC). Thus, the ReSHEALience consortium establishes the following definition: “strain hardening (fiber reinforced) cementitious material with functionalizing micro- and nanoscale constituents (alumina nanofibers, cellulose nanofibers/crystals, crystalline admixtures, especially added to obtain a high durability in the cracked state” [8].

Other definitions refer to a compressive strength of at least 120 MPa [9] without the use of fibers.

In this material, composed of a cement paste, aggregates and additions with a high level of packing, coarse aggregates are not usually utilized (the maximum size is usually between 150 and 600 µm). Although, with the increase in experience and optimization of the densification of the design, aggregates with a maximum size larger than 8 mm have been included (especially basaltic and quartzite aggregates) [10]. The water/cement (*w/c*) ratios are lower than 0.25, having been obtained through use of highly effective additives that confer self-compacting concrete rheological properties.

An adapted numerical analytical method has been adopted by using the polydisperse packing model. The aim is to achieve the highest compactness of the aggregates, cement and additions. We adapted this model to take into account the compactness and the effectiveness of the nanoadditives from the physical (compactness) and chemical (contribution to the C-S-H gels) points of view. Concluding that maximum compactness does not always lead to maximum strength. The necessary data for the characterization of the aggregates, cement and nanoadditives have been obtained through laboratory tests: dry and wet compactness, density, granulometry and water absorption.

Therefore, the significance of this research entails consideration of combining packing models, focusing on physical optimization of components with the optimal percentages and, from the chemical point of view, inclusion of nanoadditives that give the best results both in terms of strength and of durability.

## 2. Theoretical Background: Packing Models

The priorities in the design of UHPC involve achieving a high density and packing of particles. The high degree of compactness of the fine particles reduces the demand for water for the same workability, thus achieving a decrease in the water/fine (*w/f*) ratio [11]. The range of UHPC concrete considers use of additions and nanomaterials, such as silica fume, fly ash, microsilica and metakaolin, with sizes of 120 nm, or nanosilica, with a maximum size of 5 nm. The specific surface area (SSA) is inversely proportional to the particle size. The high specific surface area and the size of these materials allow them to occupy the interstices between the cementitious paste and the aggregate, thus favoring packing.

In 1616, Kepler showed that in an ordered hexagonal sphere closure (HCP) of the same diameter, the theoretical maximum packing is 74.05% ($\frac{\pi }{3\cdot \sqrt{2}}$), which is the same value for a face-centered cubic (FCC) structure. For a HCP structure, and with random packing, this would be 64% for spheres of the same diameter [12]. 

Hence, in order to increase the packing level, it is necessary to use spheres of lower diameters capable of filling the uncovered voids [10,13]. Horsfield established the maximum packing rates for five different particles for the hexagonal close [14], as can be seen in Table 1.

Packing methods can be divided into two groups: continuous and discrete (binary or multicomponent) with or without interaction. The first one is based on the creation of a continuous particle size curve, using decreasing particle sizes. There are several models referred to as continuous. Initial models such as those provided by Fuller and Thompson (1907) established a theoretical curve that relates the maximum size of the aggregate (*D*) and the sieve of the series used (*d*) with the variant of Gessner looking for a parabola [15].
(2)$$y\left(d\right)={\left(\frac{d}{D}\right)}^{r}$$

The Andreasen and Andersen model (A&A, A&A M) [16] changes the parameter of the Gessner parabola from *r* = 0.5 to *r* = 0.33–0.5 to achieve a higher degree of packing. In UHPC, a good packing level has been achieved with values of *r* = 0.23 [1]. The Fuller and Thompson model, modified by Gessner, does not establish an adequate densification for materials with particles lower than 250 µm, obtaining mixtures that are poor in cement content [17]. Conversely, the aforementioned A&A model establishes a curve up to the value of zero, with it not being limited to a minimum necessary particle size (*D_min_*). The modification of the A&A model (A&A M), established by Funk and Dinger [18], introduces the graduation of the curve and considers the minimum particle size (see Figure 1).
(3)$$y\left(d\right)=\left(\frac{d^{r}-D_{min}^{r}}{D^{r}-D_{min}^{r}}\right)$$

However, the aforementioned models are not always adapted to real systems, given that the particle size continuity of each component to be mixed is not guaranteed. The optimum packing level for a UHPC is not always reached, although the modification of the A&A method has been used in research for the manufacture of UHPC and ultra-high-performance and fiber-reinforced concrete (UHPFRC) [1,19]. These limitations can be addressed with the second group of packing models—that is to say, discrete packing models. This entails establishing particle systems where at least one is dominant, and which guarantees the solid continuity of the granular body [20]. The rest of the particles are packaged around the skeleton of the dominant class [13].

The first discrete packing model, which is valid for spheres, was Furnas’ binary model (1929), which, a year later, Westman and Hugill applied to multicomponent models [13,21,22]. In both models, there are two conditions: the first is that there is no interaction between the particles, and the second is that some particles enter the gaps left by the other particles. Furnas initially related the void index of the mixture (*ε*) with the void index of the components (*ε*_1_, *ε*_2_) and the volume fraction of component 1 in the mixture (*S_v_*_1_). By choosing the higher value for Equations (4) and (5), this may be better understood through examination of Figure 2.
(4)$$\varepsilon =1-\frac{1-\varepsilon_{1}}{S_{v1}}$$
(5)$$\varepsilon =1-\frac{1-\varepsilon_{2}}{1-S_{v1}\varepsilon_{2}}$$

Westman and Hugill identified (see Figure 3) the apparent volume *Va*, the inverse of compactness “*c*”, as the volume necessary to obtain an absolute volume equal to unity, establishing a pore index “*e*” that meets *V_a_* = *e* + 1 = 1/*c*, with *a*_1_ and *a*_2_ being the apparent volumes of the coarse and fine particles, respectively, and “*y*_1_”, “*y*_2_” being the volume fractions of each class. In Figure 3, the lines “*a*” and “*b*” are represented, respectively, as the absolute volumes of the coarse and fine particles. It should also be noted that lines “*e*” and “*d*” denote, respectively, the apparent volumes of the fine and coarse particles. Line “*c*” represents the sum of absolute volumes of all particles equal to a value of one. Line “*f*” represents the binary mixture, establishing a turning point when the void index is lower and compactness is at a maximum. This line is composed of line “*d*” for coarse particles and line “*f*”. Therefore, should the mixture with the coarse particles be dominant, Equation (6) would be obtained and for mixtures with dominant fine particles, Equation (7) would be obtained. The best mixture corresponds to the highest value of both such equations or the lowest value of Equations (10) and (11), where “*c*” is the compactness. *α*_1_ and *α*_2_ correspond to the real compactness of each class and Φ_1_ and Φ_2_ are the volumes occupied by each class in the mixed volume.
*v_a_*_1_ = *a*_1_*y*_1_ = *y*_1_/*α*_1_ = (1 − *y*_2_)(1 + *e*_1_)(6)
*v_a_*_2_ = *y*_1_ + *a*_2_*y*_2_ = *y*_1_ + *y*_2_/*α*_2_ = 1 + *e*_2_*y*_2_(7)
(8)$$y_{1}=\frac{\Phi_{1}}{\Phi_{1}+\Phi_{2}}$$
(9)$$y_{2}=\frac{\Phi_{2}}{\Phi_{1}+\Phi_{2}}$$
(10)$$c_{1}=\alpha_{1}+\Phi_{2}=\frac{\alpha_{1}}{y_{1}}=\frac{\alpha_{2}}{1-y_{2}}$$
(11)$$c_{2}=\alpha_{2}+\left(1-\alpha_{2}\right)\Phi_{1}=\frac{\alpha_{2}}{1-(1-\alpha_{2})y_{1}}$$

Authors such as Ben-Aïm [24] (1970), Sotovall et al. [25], and De Larrard et al. [26] (1986) examined the two most important restrictions to packing. Another study, which entailed proposing the wall and loosening effect, was provided by Caquot in 1937. The wall effect occurs when the predominant fines or container displace them locally. The loosening effect takes place when the size of the fines is greater than the voids they occupy, distancing the thicknesses. The relationship between particle size (*d*_1_*/d*_2_) is fundamental to these effects.

Sotovall and De Larrard, also based on Mooney’s model (1950) for predicting the viscosity of monodisperse particle suspensions in a liquid medium, created a method of packing by searching for the proportion necessary to achieve an infinite viscosity. They named it the linear packing density model (LPDM) [27], where the wall effect and loosening interactions were established. However, given that it is not possible to represent the random behaviour of the real packing, the solid suspension model (SSM) was established later, where viscosity is delimited and becomes finite (introducing the interactions between virtual and real compactness).

In the LPDM model, as shown in Equation (12), the packing of a “*t*” particle is established and considers the loosening and wall effects. Thus, Equations (13) and (14) represent the loosening and wall effects, respectively, of the mixture and depend on the binary relations or interactions between the particle sizes (*z; d*_1_*/d*_2_). *y*(*t*) is the volume fraction of size “*t*” in the mixture and, taking into account Equation (15) $\left(\int_{d}^{D}y\left(x\right)dx=1\right)$, where *d* and *D* are the minimum and maximum particle sizes, respectively, *β*(*t*) is the virtual packing density and *α*(*t*) is the real packing density.

In the SSM model, the starting point is Mooney’s viscosity model [28], which relates the solid content *φ*, ordered at random, and the relative viscosity *η_r_* according to Equation (17) (a model used again by several researchers from the 1980s onwards with the advance of additives and the need to link workability and compactness). Thus, taking the HCP structure as an example, where the packing density of the ordered spheres is 0.74 and such density of the unordered form is 0.64, a relative viscosity of *η_r_* = 1.36 × 10^5^ may be obtained, called reference viscosity *η_r_^ref^*. Equation (17) establishes the variation of viscosity *η_r_*, for a particle size of “*t*”, between *D* and *d*, depending on the virtual *β*(*t*) and real packing *α*(*t*).

Equation (16) establishes the virtual packing density *β*(*t*) as a function of the real packing *α*(*t*) through the adapted Equation (17) of Mooney’s viscosity. Equation (18) represents the ratio of “*yi*” volumetric fractions and virtual packing in a system, with no longer binary but *N* particles.
(12)$$\begin{array}{c} c\left(t\right)=\frac{\alpha \left(t\right)}{1-\int_{Dmin}^{t}y\left(x\right)f\left(\frac{x}{t}\right)dx-\left[1-\alpha \left(t\right)\right]\cdot \int_{t}^{D}y\left(x\right)g\left(\frac{t}{x}\right)dx};\\ c=\min\left(c\left(t\right)\right);\;y\left(t\right)>0 \end{array}$$
(13)$$f\left(z\right)=0.7\left(1-z\right)+0.3\;{\left(1-z\right)}^{12}$$
(14)$$g\left(z\right)={\left(1-z\right)}^{1.3}$$
(15)$$\overset{D}{\underset{d}{\int }}y\left(x\right)dx=1$$
(16)$$c\left(t\right)=\frac{\beta \left(t\right)}{1-\int_{d}^{t}y\left(x\right)f\left(\frac{x}{t}\right)dx-\left[1-\beta \left(t\right)\right]\cdot \int_{t}^{D}y\left(x\right)g\left(\frac{t}{x}\right)dx}$$
(17)$$\begin{array}{c} \eta_{r}^{ref}=\exp\left(\frac{2.5}{\frac{1}{\alpha \left(t\right)}-\frac{1}{\beta \left(t\right)}}\right);d\leq t\leq D;\eta_{r}^{ref}=\exp\int_{d}^{D}\left(\frac{2.5\cdot y\left(t\right)}{\frac{1}{c}-\frac{1}{c\left(t\right)}}\right);\\ \eta_{r}=\exp\left(\frac{2.5}{\frac{1}{\Phi }-\frac{1}{\beta }}\right)=\eta_{r}^{ref}\\ \mathrm{HCP}:\;\Phi =0.64;\;\beta =0.74;\;\eta_{r}^{ref}=1.36\times 10^{5} \end{array}$$
(18)$$\frac{1}{\beta \left(t\right)}=\overset{N}{\underset{1}{\sum }}\frac{y_{i}\left(t\right)=1}{\beta_{i}\left(t\right)}$$

Yu and Standish [29] (1987) developed a model akin to that provided by Sotovall et al. from the initial model offered by Westman. 

In 1999, De Larrard et al. developed the compressible packing model (MEC), which introduces the compaction index *K*, representative of the energy supplied in the compaction and the type of stacking—that is to say, the process of building the stack. This led to a third-generation model being established where, in addition to taking into account the real and virtual packing of the mixtures, it introduced packing levels as a function of the compacting energy of the relationship between both packing densities [23].

This model establishes the calculation of the virtual compactness *γ* based on the characteristics of each particle type in an ordered mixture, proportion *y_i_* and unit virtual compactness *β_i_*. Subsequently, the method calculates the real compactness *φ* (*α_i_* of each particle) ordered randomly. The compaction index *K* relates each compact; the real compactness of the mixture *φ* grows with the value of *K*. Thus, for each *K* index, a real maximum compactness *φ* determined in the mixture will be achieved.

For the general case with interaction in a polydisperse system, the MEC starts from a ternary mixture, where the dominant class with grain size *d*_2_ suffers the effect of loosening, *a_ij_*, by the class with grain size *d*_3_, with *d*_2_ > *d*_3_ and the wall effect, *b_ij_*, by the grain class *d_1_* with *d*_1_ > *d*_2_. Equation (19) establishes the virtual compactness of a mixture where class *i* is dominant.
(19)$$\gamma_{i}=\frac{\beta_{i}}{1-\sum_{j=1}^{i-1}\left[1-\beta_{i}+b_{ij}\beta_{i}\left(1-\frac{1}{\beta_{j}}\right)\right]y_{j}-\sum_{j=i+1}^{n}\left[1-a_{ij}\beta_{i}/\beta_{j}\right]y_{j}}$$

Then, real compactness is established. As indicated in the previous paragraphs, the real compactness *φ* responds to a packing of the particles by means of a determined compacting method. By establishing that *φ* < *γ*, the method identifies a relationship between virtual and real packing, through the compaction index *K*. An increase in the efficiency of the compaction method leads to a subsequent increase in real compactness. Each compaction method (discharge, crushing, vibration, vibration + pressure, as shown in Table 2) has a specific value of *K* [14,30] (Equation (21)). The compaction index of grain class “*i*”, *K_i_,* will be related to the actual compactness (volume of the solid), *φ_i_*, of that grain class and to the maximum actual compactness (volume of the solid), *φ_i_**, which grain class “*i*” will have in the presence of other grain classes. When class “*i*” is dominant in the mixture *φ_i_** = *φ_i_* (*K* = *∞*), the *H* function value is between 0 and 1, and ratio variation *φ_i_/φ_i_** of Equation (21) exhibits a trend to the value of zero for the minimum compaction index and to the value of one when the compaction index is at the maximum value (*K* = ∞). In Equation (23), the compaction index is established as the relationship between the known fractions of each class *y_i_*, the compactness *β_i_*, *γ_i_* and the real compactness of the mixture. The implicit equation is in *K*, given that the rest of parameters are known or calculated according to their placement, proportion and interaction [30]. According to Equation (23), for each compactness value *φ* there is a value of *K*. For a packing of particles of the same size, Equation (24) would be obtained, which is implicit in *β_i_*.
(20)$$K=\overset{n}{\underset{i=1}{\sum }}K_{i}$$
(21)$$K_{i}=H\left(\frac{\Phi_{i}}{\Phi_{i}^{*}}\right)$$
(22)$$K=\overset{n}{\underset{i=1}{\sum }}\left(\frac{\frac{\Phi_{i}}{\Phi_{i}^{*}}}{1-\frac{\Phi_{i}}{\Phi_{i}^{*}}}\right)$$
(23)K=∑i=1nKi=∑i=1nyiβi1Φ−1γi
(24)$$K=\frac{1}{\frac{\beta_{i}}{\Phi_{i}}-1};\;\beta_{i}=\frac{1+K}{K}\cdot \Phi_{i}$$

The first approximation to the wall effect was made by Caquot in 1937, indicating that the reduction in compactness of the smaller grains with size *d*_2_ around the thicker grains with size *d*_1_ results in a variation of volume proportional to the surface of the interface [23]. Caquot identified a linear relationship between 0 and 1 for the wall coefficient, *b*_12_ = *x*; (*x* = *d*_2_/*d*_1_). Authors such as Ben-Aïm established a disturbance zone in the contact between coarse and fine particles where a reduction in compactness occurred, depending on the level of insertion of the fine grains in the disturbance zone, an approximation that can be established is *b*_12_ = 2*x*. Therefore, for grain sizes with *d*_1_ = *d*_2_, the wall coefficient does not trend toward the value of one. Dodds provided a model for calculating the wall coefficient based on the theoretical model of hexagonal-close packing (HCP), with *β* = 0.74, where the tendency of the wall effect for *d*_1_ = *d*_2_ tends toward the value of one, obtaining values greater than one for values of *x* between 0.6 < *x* < 1. However, this model establishes a maximum packing, something that does not occur in reality, with the voids left being filled by particles that also meet the condition of full contact. The function that represents the wall coefficient indicated in the MLC was obtained empirically by means of the previously mentioned theoretical models; Equation (25) represents the wall coefficient of the MLC. This function was later adjusted until Equation (26) was obtained in the MEC. Figure 4 represents the function of the wall effect coefficient, tending to the value of one for *d*_1_ = *d*_2_.
(25)$$b\left(x\right)=1-{\left(1-x\right)}^{1.6}$$
(26)$$b\left(x\right)=1-{\left(1-x\right)}^{1.5}$$

In the MLC model, Sotovall [25] offers a hypothesis to establish the alienation effect in a binary mixture. It first indicates that the void left in space (3D) by four contacting spheres of diameter *d*_1_ may be occupied by a sphere of diameter *d*_2_. This, in turn, will be tangent to the rest of the spheres and therefore will not cause separation or a reduction in compactness. The critical diameter ratio, between the spheres of different diameters, would be *x*_0_ = 0.224 (*x* ≈ 0.2), (*x*_0_ = 0.154 for three spheres in one plane), as depicted in Figure 5. It also shows that the insertion of spheres with a diameter *d*_2_ greater than the value $x_{0}\cdot d_{1}$ leads to a decrease in the point compactness of particles with a diameter *d_1_*, from *β*_1_ to *β′*_1_. The voids left will be filled with small particles whose proportions *β** are linearly related to the ratio of diameters *x*. In addition, when *x*→1, compactness with dominant fines or coarse is equal and their volume fraction is equal to 0.5.

The loosening function embodied in the MLC is set out in the following equations, indicating that for a lower ratio of *x*_0_ < 0.2 the value of the coefficient is zero. Figure 6 represents the function of Equation (27), for random packing (*β* = 0.64) and HCP (*β* = 0.74), establishing an upper limit of 1 and a considerable increase in the distance coefficient between 0.2 < *x* < 0.3.
(27)a(x)=(1−(x0x)3)(1−x03)·((1−β−3x031−x03(1−x)+1)· x≥x0
(28)$$a\left(x\right)=0\;x<x_{0}$$

Larrard [23,26,30] provided a function of the loosening coefficient according to empirical measurements. In this case, the hypothesis of *a*(0) = 0 and *a*(1) = 1 is maintained, though no null value is established for diameter ratios of less than 0.2, as indicated by Sotovall (though the horizontal tangent is established at 1 and 0). Equation (29) shows the function of the loosening coefficient proposed by Larrad. Other authors, such as Yu et al., established a function of the loosening coefficient with higher values for lower size ratios [23,31], with a similar trend to that indicated by De Larrard. Equations (29) and (30) establish, respectively, the functions of De Larrard and Yu.

In later research, Sotovall et al. [23] carried out empirical studies on samples of rolled and crushed aggregates, identifying a new function of the loosening coefficient, indicated in Equation (31). It should be noted that there is no horizontal tangent at 1 and a vertical tangent is established at 0, and that the values of *a*(0) and *a*(1), continue to prevail. This function was modified for other experimental cases provided by Lecomte et al., as shown in Equation (32). Lastly, in the De Larrard SCM method, a new function was established for the loosening coefficient, indicated in Equation (33). Figure 7 shows the functions of the loosening coefficient according to these studies.
(29)$$a\left(x\right)=1-{\left(1-x\right)}^{3.1}-3.1\cdot x\cdot {\left(1-x\right)}^{2.9}$$
(30)$$a\left(x\right)=1-{\left(1-x\right)}^{3.33}-2.81\cdot x\cdot {\left(1-x\right)}^{2.77}$$
(31)$$a\left(x\right)=\sqrt{x}$$
(32)$$a\left(x\right)=x^{0.414}$$
(33)$$a\left(x\right)=\sqrt{1-{\left(1-x\right)}^{1.02}}$$

## 3. Effect of Hydration of Nanoadditives

It is clear that the process of establishing a high compactness in the granular skeleton is a guarantee of obtaining a high-strength UHPC. In addition to this, the hydration products formed by the addition of nanoadditives also lead to an increase in strength that does not depend on the degree of compactness [1,32]. When cement is hydrated without additives, this only occurs at the surface of the grain, meaning that the hydrated products grow and settle around it. This process prevents or reduces the ion transfer between the unhydrated cement particles and the surrounding solution, limiting the formation of a denser matrix of C-S-H [1]. The use of nanosilica will, in addition to increasing the degree of compactness, produce an early pozzolanic reaction on the silica surface and the creation of larger C-S-H gel layers and higher crystallization, which will lead to a less porous network and more resistant concrete. Low percentages of nanosilica (some published authors suggest less than 2%) have been shown to provide high values of strength. Nanofillers improve the bond between aggregates and mortars, thus obtaining a stronger structure [33].

In this work, it has been observed that 1.5% nanosilica in the presence of other additions does not lead to a significant increase in the concrete strength. Nevertheless, a high content of nanosilica (close to 5%) creates a greater network of C-S-H, with nanosilica conferring greater viscosity to the paste, which can trap air in the system and increase porosity. Studies by Yu et al. [1] showed that values higher than 4% for nanosilica caused an increase in the porosity of UHPC. The use of metakaolin affects the hydration kinetics and, for certain percentages of metakaolin, a greater quantity of C-S-H can be obtained [2]. In Kunther et al.’s research on the hydration kinetics of cements with different percentages of metakaolin substitution, the authors achieved an initial increase in the amount of C-S-H phases and a decrease in the portlandite content, accompanied by an increase in the amounts of calcium aluminate hydrate phases. This increase in the C-S-H phase was also seen with the joint use of nanosilica and metacaolin at 90 days in cement pastes in research carried out by Silva Andrade et al. [34] Therefore, there was a synergistic effect in the use of the two additions. The packing models did not consider the optimal percentages as a function of hydration, which should always be present, as indicated above.

## 4. Experimental Campaign: Material Characterization and Results

### 4.1. Material Characterization

Initially, three aggregates with sizes *S*_3_ (0.5–1.6 mm), *S*_2_ (0–1 mm), *S*_1_ (0–0.5 mm) were used in the design of the granular skeleton. The experimental compactness, dry compactness and *C_i_* of these aggregates were determined, with a compaction index of *K* = 9 that provided values of *c*_*S*3_ = 0.57, *c*_*S*2_ = 0.62 and *c*_*S*1_ = 0.56. The method used is that provided in LPC test no. 61 [35], with the variation through vibration being that indicated by Sedran [36]. By means of the MEC, the maximum compactness reached was analyzed as a function of the percentage of each aggregate. A greater degree of compactness was achieved in the mixture of aggregates *S*_3_-*S*_1_ with 60% of aggregate *S*_3_, achieving a compactness *φ*_*S*3−*S*1_ = 0.70. For a better understanding of this procedure, see Figure 8, Figure 9 and Figure 10. In Figure 9, it can be seen that the use of the sands *S*_3_ and *S*_1_ present the highest compactness. This value is higher than that obtained with the mixture of the three sands.

In order to increase compactness, additions with sizes lower than 100 µm were used (including cement). The additions used in the models were limestone filler, metakaolin, silica fume, and nanosilica. The size of them can be seen in Figure 10.

In order to identify the compactness of these materials, the water demand test (*K* = 6.7) and wet compactness (in accordance with standard EN 196-3 [37] and Sedran studies [38]) were used to compute Equation (34), where *c* is the compactness, *m_H*20*_* is the mass of the water used, *ρ* is the density of the sample after drying, and *m* is the dry mass.
(34)$$c=\frac{1}{1+\rho \cdot \frac{m_{H2O}}{m}}$$

Such a test is required to consider the dispersing or deflocculating effect of the additives which lead to an increase or change in the compactness of the tested material [36,39]. The additive used was a high-performance superplasticizer Sika^®^ ViscoCrete^®^-20 HE. The consequent additive ratio was established up to the saturation point of the additive, determined by means of a Marsh cone according to ASTM C-939 [40]. The flow time, and therefore the saturation point, varied depending on the addition used. The flow times of CEM I 52.5 R and of the silica fume cement slurry were determined in order to examine variation with and without addition, extrapolated to the behavior with other additions. The results obtained indicated, for 100% cement, that the saturation point is established at 1.2% in weight of additive for the mixture of 80% cement and 20% silica fume and the saturation point at 3% in weight of the additive. This can be observed in Figure 11. It can be seen how the mixture of cement with nanoaddition requires a larger amount of additive to achieve the same rheology.

Once the saturation points of the additive were determined, the wet compactness tests were carried out, verifying the variation of compactness with the percentage of additive. Thus, for filler and metakaolin, maximum values of compactness were obtained—*c_FILL_* = 0.64, *c_FILL_* = 0.63 for nanosilica and cement values of *c_cem_* = 0.62, *c_Ns_* = 0.55 for percentages in weight of additive of 1.2%. Silica fume compactness was *c_SF_* = 0.60 with 3% wt. of superplasticizer additive. Figure 12 shows the experimental compactness of additions according to the percentage of additive. The importance of the additive in obtaining the compactness values can be seen in this figure. Low additive contents lead to lower compactness values in the materials to be mixed. This directly affects the mixed compactness of the ensemble.

In addition to the influence of the additive on the compactness, the packing method was implemented by using the general equation for a polydisperse system of particles, both active and inert. It was adapted for specific percentages of nanoadditives. The numerical analysis established in this method allows the compactness of the solid mixture to be achieved. 

### 4.2. Results and Discussion

Designs were established with sand S_3_ and sand S_1_ and with a maximum of three additions. The percentage volume of cement was established as between 35.2% and 43.3%. The compaction index was established at *K* = 6.45, which was slightly lower than the wet compactness value and close to the *K* = 6.7–7 value used in self-compacting concrete [36]. The maximum compactness was achieved with the combination of three additions and nanoadditions, filler or metakaolin + silica fume + nanosilica (starting with value five in Table 3). The maximum compactness value, *φ* = 0.7899, was obtained with the combination of filler + silica fume + nanosilica, a value that was close to that achieved with the mixture of metakaolin + silica fume + nanosilica (*φ* = 0.7881) with both mixtures having the same cement content. It can be seen that the use of a higher number of additions with smaller particle sizes achieved higher compactness. In summary, all the models described above seek to increase compactness as the main premise for achieving higher strength. However, as will be seen in the following paragraphs, this should not be the only premise to search for.

As indicated, the packing models are not enough to obtain the results and properties sought after in UHPC. The search for maximum compactness is not a guarantee of success. It is necessary to use an adequate combination in the additions in order to complete the hydration, since the improvement of the additives allows a reduction in the *w*/*c* ratio up to values of 0.18. Studies by authors such as Rong, Xiao and Wang [41] verified the effect of silica fume on the hydration processes and microstructure of UHPC concrete, concluding that at low water/binder ratios silica fume dominated the hydration processes. Rong et al. [42] carried out a replacement of cement percentages by silica fume in UHPC mixtures. The researchers concluded that percentages of up to 3% of silica fume content led to an increase in compressive and flexural strengths of UHPC. Higher values create the opposite effect, due to an agglomeration of silica fume particles. The addition of nanosilica in percentages of up to 4% by weight of SiO2 cement accelerates the formation of C-S-H compounds, achieving higher strengths at all ages [43,44,45]. Land and Stephan [46] determined that there was an increase in the heat of hydration with the increase in the surface area. With percentages of between 3% and 5% in weight, increases in compressive strength of between 14% and 17% were obtained with a decrease in the permeability of the concrete. Other authors, such as Brouwers et al. [17] and Yu et al. [19], have established values of up to 3.76% in nSi as optimal values for the improvement of mechanical properties. However, Senff et al. [47] did not achieve large increases in the compressive strengths of mortars and pastes with the addition of nanoparticles (SiO_2_ and TiO_2_), though they did achieve changes in rheology with increases in plastic viscosity. Authors such as Oertel et al. [48] achieved an increase in compressive strength at ages of more than seven days with the addition of nSi in UHPC, though no clear relationship was established between the level of packing acquired, hydration and development of microstructure in the mixtures and the expected strengths. The use of metakaolin improves the workability times of the concrete and also needs a lower quantity of superplasticizer to achieve the same consistency. Tafraoui et al. [49] achieved an improvement in the bending strength of UHPC treated at 90 and 150 °C. However, El Gamal et al. [50] made mixtures combining metakaolin and silica fume with improved results in strength than only using metakaolin. Morsy et al. [51] achieved increases in flexural strength in mortars mixed with metakaolin (up to 7.5% in weight of mixture) and silica fume that, with the addition of the latter for pastes with the same *w*/*c* ratio, decreased by between 7.5% and 10% in terms of strength. 

Table 4 shows the percentage by volume of the components of each mixture, as well as the compaction index and compactness achieved. Additionally, it presents the mixing times and compressive strengths at 7 and 28 days.

The materials listed in Table 4 were mixed in a planetary mixer with a vertical axe. The procedure was to first mix the aggregates for 10 s, then the cement and the nanoadditives were introduced and mixed for 10 s. Subsequently, 50% of the water, mixed with the additive, was added and mixed for 10 s. The remaining 50% of the water was then added. The mixing and current intensity measurements then began until a stable value of the amperemeter was achieved.

Table 4 sets out the results of the mixing times with *w*/*c* = 0.18. The mixing times increased with the number of additions. Mixtures 1–3 had equal times, with the same occurring with Mixtures 5–7. Mixture 8 had a longer mixing time than mixtures with the same number of additions due to the increase in filler content even though there was a reduction in silica and nanosilica fume content. Mixture 9, with the same silica and nanosilica fume contents as Mixture 8, experienced a reduction in mixing time as it was designed with metakaolin instead of filler. These mixing times were needed to ensure the consistency and fluidity of self-compacting concrete.

Figure 13 shows the electric intensity experienced by the engine of the mixer during the mixing process compared with the elapsed time. This allows an indirect measure of the consistency of the concrete to be established. In addition to setting the most appropriate mixing times, in this figure, the mixing times and engine intensity have been transferred for a mixture with two additions, Mixture 1, and with three additions, Mixture 5. This analysis should be carried out with the same mixer and with the same volume of mix. The intensity vs. mixing-time curve is usually divided into three zones. In the first zone, the curve experienced an increase in intensity vs. time given; in said time, the additive and the mixing water did not produce a substantial modification of the rheology of the mixture. In the second zone, the curve has an inflection point where the intensity is at its maximum value, from which such intensity is reduced as the additive and the mixing water modify the rheology of the mixture, making it more fluid. The third zone of the curve tends to be horizontal, achieving maximum fluidity of the mixture and determining the mixing time. It can be seen that Mixture 5 has a higher peak intensity than Mixture 1 due to the greater number of additions, which makes it more difficult for the additive and the water to modify the rheology, although this occurs earlier than in Mixture 1. The second and third zones are similar across the two mixtures, with Mixture 5 taking a few seconds more than Mixture 1. The control of the rheology of the mixture was achieved by measuring the current intensity consumed by the mixer. It also predicts when the mixture has a stable rheology and is suitable for pouring into the mold.

Table 4 also shows the results obtained for compressive strength in a cubic specimen. For Mixtures 1–3 with two additions, with the same cement content and compactness, a seven-day average strength of 82.7 MPa and a deviation of 3.45 MPa were obtained. At 28 days the average strength achieved was 119.5 MPa with a greater deviation of 4.42 MPa. Mixtures 5–9 were made with the same percentage of cement, higher than that used in Mixtures 1–3 and three additions. With different values of compactness, though higher than those obtained with two additions, Mixtures 5–6 exhibited lower compression strength values than Mixtures 7–9, even though they were more compact (even Mixture 5 experienced a drop in strength at 28 days compared with seven days). If Mixtures 5 and 6 are compared with Mixture 9, it is clear that the metakaolin used confers a greater strength to compression with less compactness.

Figure 14 sets out the results provide in the table above. By analyzing the results of the group of Mixtures 1–3 with two additions, with a lower cement content and less compactness than the group of Mixtures 5–9 with three additions, with a higher cement content and greater compactness, it may be determined that greater compactness in the mixture does not lead to greater strength, although mixtures with greater compactness may have a greater durability.

Figure 15 compares the compression strength at seven and 28 days with the achieved compactness. At seven days, there is a direct proportional relationship between compactness and strength achieved, also considering that the mixes with greater compactness have higher cement contents. However, this relationship is reversed with the results at 28 days, with higher compressive strengths being achieved in mixes with two additions (with less compactness and lower cement content).

Figure 16a shows a SEM image of the concrete obtained with metakaolin, nanosilica and silica fume (mix #9), a dense and closed structure of hydrated products can be observed. In addition, the nanoaddition may be observed on the cement hydrated products—more specifically, C-S-H gel. The image on the right shows, for comparison, the hydrated cement without nanoadditions. 

In addition, three mixes were analyzed by Differential Thermal Analysis (TGA/DTA). TGA/DTA profiles, according to ASTM E 1131 [52], were registered by using Setaram Labsys EVO TGA apparatus. A quantity of 50–70 mg of each sample was heated at 10 °C/min up to 1100 °C. In order to stop the hydration of the samples at the age analyzed (28 days), the concrete was placed in a vacuum for 30 min and then in isopropanol for 24 h, followed with 24 h in a stove at 60 °C. The sample was stored in a stove at 40 °C until its characterization was carried out. 

Plotting the derivative of the weight loss or what is the same the speed of weight loss (dTG) vs. temperature is more useful than the representation of the weight loss (TG) vs. temperature as it allows, in a way, a more clear and unequivocal identification of the dif- ferment start and end temperatures of different processes of weight loss with temperature. From the temperatures selected for the representation of dTG vs. TG, the quantification of the weight loss with the temperature associated with the different reactions that occur in the cement was carried out. Figure 17 shows the dTG vs. temperature of Mixtures 3, 7 and 8. Based on the methods proposed by Bhatty [53], Pane et al. [54] and Monteagudo et al. [55], various regions of dehydration, Ldh (region *T*_1_), dehydroxylation, Ldx (region *T*_2_) and decarbonation, Ldc (region *T*_3_), are highlighted in Figure 17.

The selection of the mixtures for TGA/DTA analysis was oriented to assess the influence of the nanosilica in the hydration of the cement. According to this, Mixture 3 (without nanosilica) and Mixtures 7 and 8 (with two contents of nanosilica, see Table 4) were selected. 

Table 5 shows the Portlandite, C-S-H gel and carbonate contents based on the TG results of Mixtures 3, 7 and 8 at the age of 28 days. The key values for comparison the influence of the nanosilica in the cement hydration are the ratio water of C-S-H gel/cement content and the water of free Portlandite/cement content. Mixture 3 shows the highest values of both ratios. This is consistent with the high degree of hydration that silica fume usually produces at the age of 28 days. Mixture 7, with the highest content of nanosilica, shows the lowest values for both parameters, indicating the lowest hydration caused by the delay produced by nanosilica in the hydration of cement at early ages. This aspect has been observed by other researchers [56] when nanosilica is very active, and the hydration products cover the cement grain and make the access of water to cement difficult. Mixture 8, with a medium content of nanosilica, shows intermediate hydration values compared to those shown for Mixtures 3 and 7.

This article has analyzed the packing models from their beginnings and evolution to the most developed models. Initially, continuous models, such as those offered by Fuller and Thompson, sought to achieve a theoretical curve or parabola by relating the size of particles to their maximum size. The development of these models was centered on the variation of the parabola parameter (Gessner’s parameter and modifications made to it by Andreasen and Andersen [16]). In addition, the relation with the minimum size, allowing a greater compactness of the mixtures but not permitting an adequate response when using particles smaller than 250 µm, was taken into account. The need to establish greater compactness in sectors other than concrete leads to the appearance of discrete models, initially binary, such as that provided by Furnas, which evolved into multicomponent models where the relationship between particles of different sizes in a given mixture could be established. Later, with the studies carried out by Ben-Aïm, Sotovall and De Larrard et al., based on previous works by Caquot and Mooney, the wall and loosening effects that occur in the mixture of particles of different sizes and their environment were established, which affect the compactness of the bulk material. Later, De Larrard also established the relationship between virtual and real compactness according to the method and index of compaction to which the mixture is subjected. The degree of compactness can be determined according to the external conditions of placement. These models make it possible to obtain multicomponent mixtures independently of their size. The need to search for high-degree packing in the UHPC makes these models the most relevant ones. The development of additives that allow the addition of water to be necessary for hydration of the particles makes such models more significant in the field of concrete. However, in addition to maximum compactness, the hydration conditions of the compounds and the contribution to the fresh and hardened states should be considered. Therefore, a design is required to respond not only to the acquisition of maximum compactness of the solid components but also the appropriate proportions of those solids regarding hydration and activity. Thus, in mixtures with a lower content of additions, lower cement content and less compactness, higher compressive strength values are obtained than in mixtures with a greater number of additions, cement content and compactness. However, the latter are expected to provide the concrete with greater durability. This was verified in mortars examined by Alonso Dominguez et al. [57] where the use of nanoadditives such as nanosilica allowed a porous network with a smaller pore diameter to be obtained. This led to an improvement in electrical resistivity and chloride migration coefficients, thus providing the mortars with greater durability, although these studies were not carried out with as great an amount of cement content and such a low *w*/*c* ratio.

## 5. Conclusions

This article analyzes different models that aimed to achieve the highest levels of compactness through an ideal curve. The pioneering work of Féret [58] determined the influence of aggregate types and their combinations on the strength of concrete. In order to fulfil Fuller’s premise, that higher compactness leads to higher strength, the evolution from binary, interacting and non-interacting models to polydisperse particle packing models with interaction has been described. However, the following conclusions state that, in addition to the initial assumption of compactness (physical aspect), the chemical effect of the additions in the mixture must be taken into account.
(1)A densification of UHPC by means of packing models in combination with the assessment of the activity of nanoadditives in the hydration processes of cement is a promising way for improving the design of UHPC. The maximum compactness of the particles, when nanoadditives are used, does not always obtain the best strength and durability of the UHPC. Consideration of the role of nanoadditions in gaining strength and durability properties is a key aspect.(2)The highest compactness of the three sands used was achieved with the use of two of them (the thickest, S_3_ and S_1_, with 60% of aggregate S_3_ achieved, through De Larrard’s packing model, a compactness of φ_S3−S1_ = 0.70). The compactness was unaffected by the use of an intermediate sand, S_2_.(3)The use of nanoadditions of various sizes permits an increase, through use of the same model, in the compactness of the mixtures. The higher degree of compactness was achieved by using three additions: limestone filler, silica fume and nanosilica (φ = 0.7899).(4)The measurements of the mixing times, by using the amperemeter of the mixer, were always longer for the mixes with three nanoadditives than for ones with two nanoadditions for the same percentages of additive.(5)The designs made with three additions and higher cement contents led to better compressive strengths at seven days than those designs made with two additions and lower cement contents. The presence of nanoadditives, such as nanosilica, improved C-S-H gel formation and thus achieved better results at earlier ages.(6)The compressive strength results for 28-day-old mixtures with two additions were good and even higher than those obtained with mixtures with three additions, in spite of exhibiting lower compactness and cement contents. A higher percentage of nanoaddition in the mixture may be inadequate and lead to the opposite effect.(7)The mixture with additions of metakaolin and nanosilica, with the latter showing slightly lower percentages than other mixtures with three additions 1.5% in volume, allows higher compressive strengths alone to be achieved compared to when using nanosilica. The synergy in the use of both components is demonstrated.

## Figures and Tables

**Figure 1 nanomaterials-11-01414-f001:**
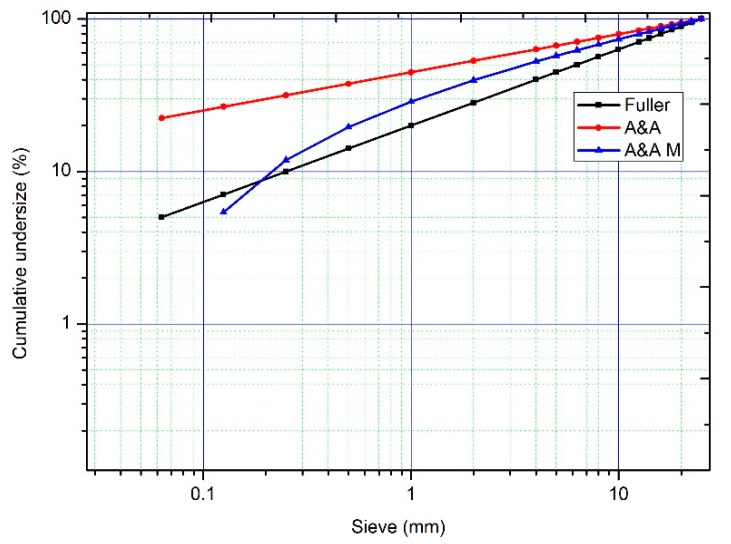
Particle size distributions: Fuller, A&A, A&A M *D* = 25 mm, *D_min_* = 63 µm.

**Figure 2 nanomaterials-11-01414-f002:**
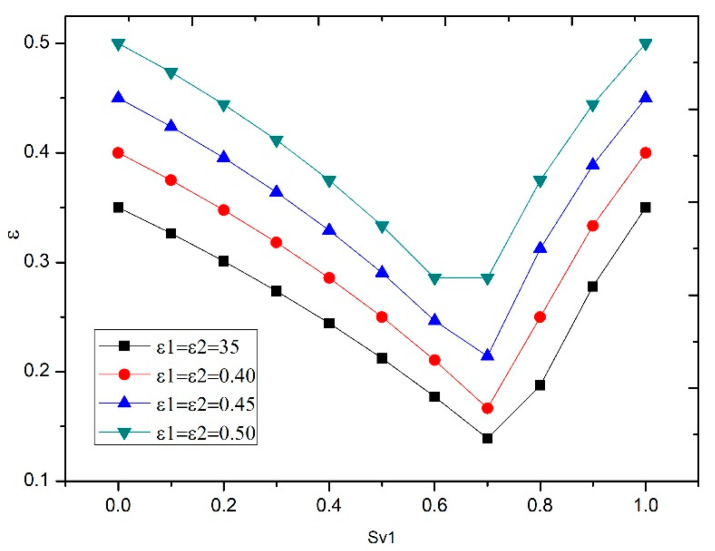
Void fraction for various fractions of component 1.

**Figure 3 nanomaterials-11-01414-f003:**
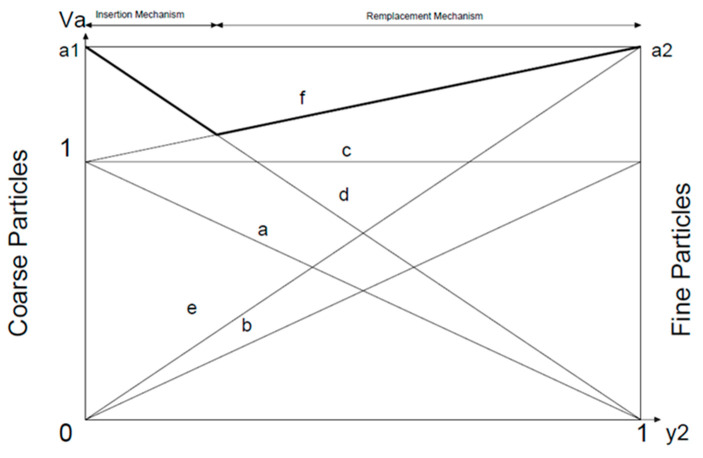
Evolution of bulk volume in binary mixture as a function of the proportion of fine particles—Westman and Hugill model, prepared by the authors based on data from [23].

**Figure 4 nanomaterials-11-01414-f004:**
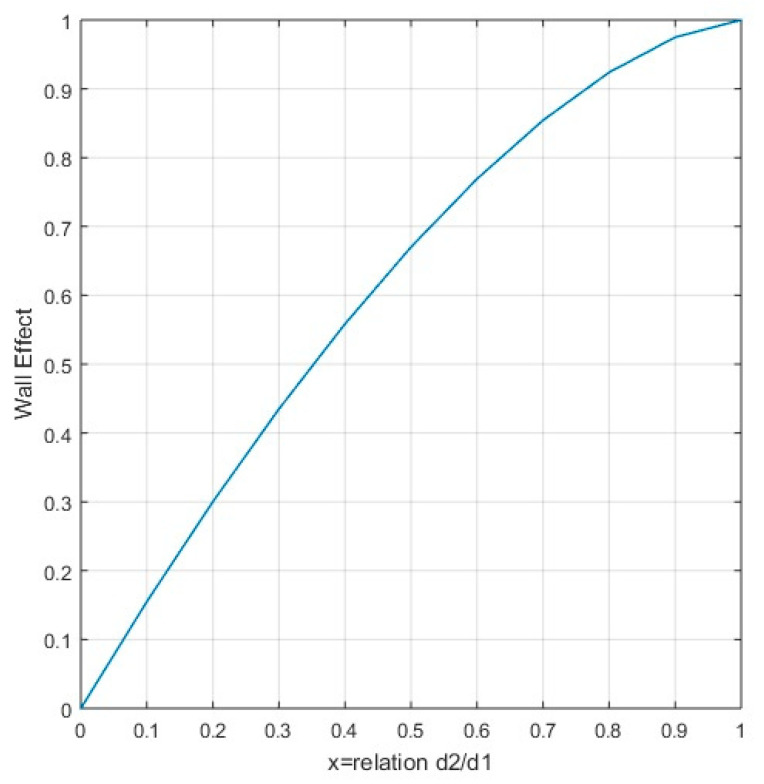
Wall effect function in MEC.

**Figure 5 nanomaterials-11-01414-f005:**
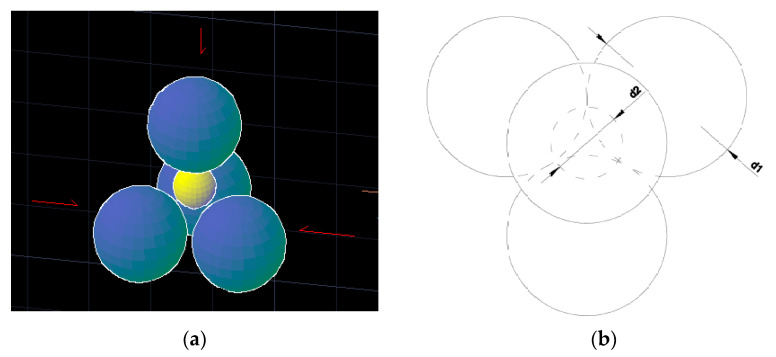
Packing of four spheres with diameter *d_1_* and *d_2_* = 0.224 *d*_1_. (**a**) 3D visualisation, and (**b**) vertical projection.

**Figure 6 nanomaterials-11-01414-f006:**
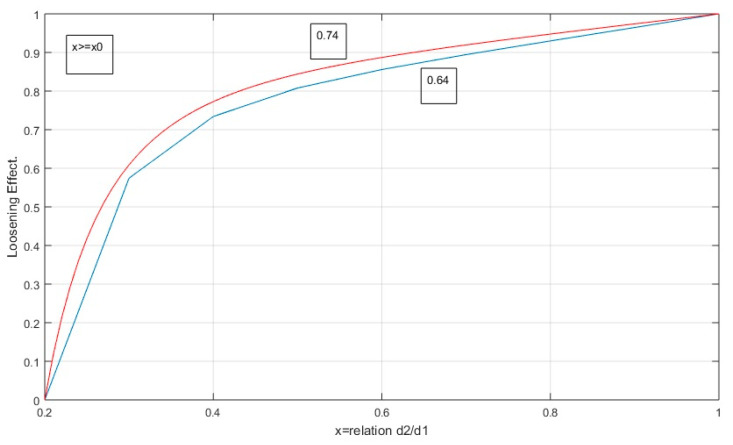
Loosening coefficient function in MLC.

**Figure 7 nanomaterials-11-01414-f007:**
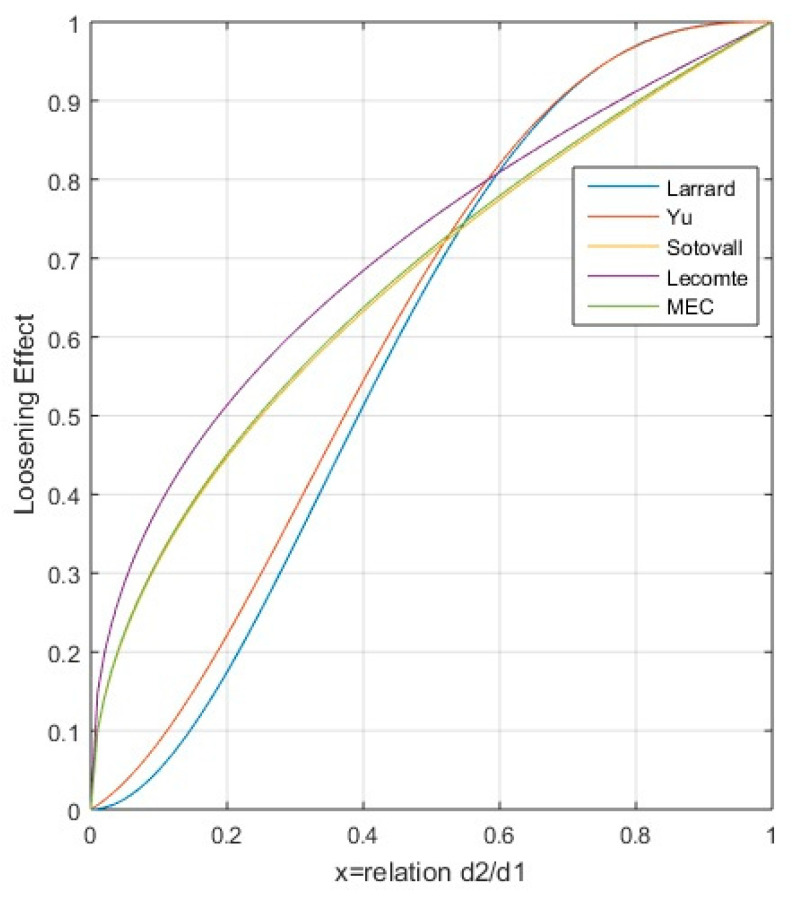
Loosening coefficient functions.

**Figure 8 nanomaterials-11-01414-f008:**
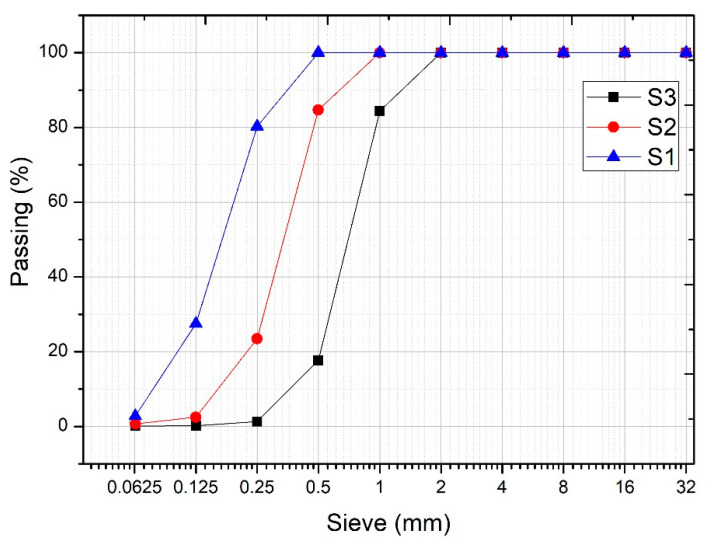
Percentage passage of the aggregates used.

**Figure 9 nanomaterials-11-01414-f009:**
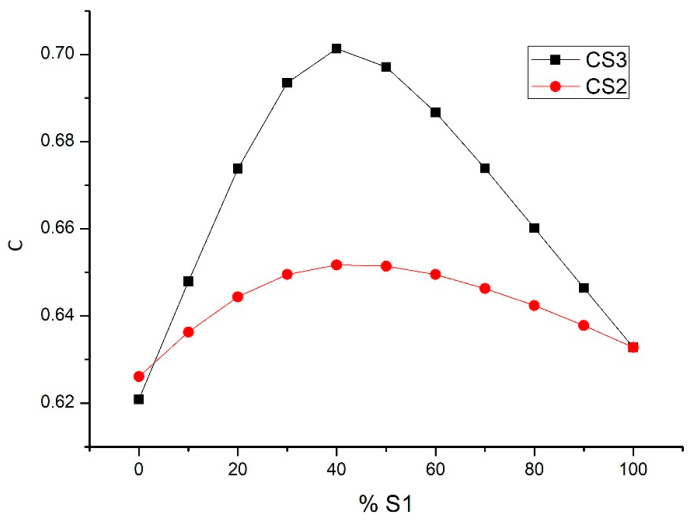
Aggregate compactness, *S*_3_ − *S*_1_, *S*_2_ − *S*_1_.

**Figure 10 nanomaterials-11-01414-f010:**
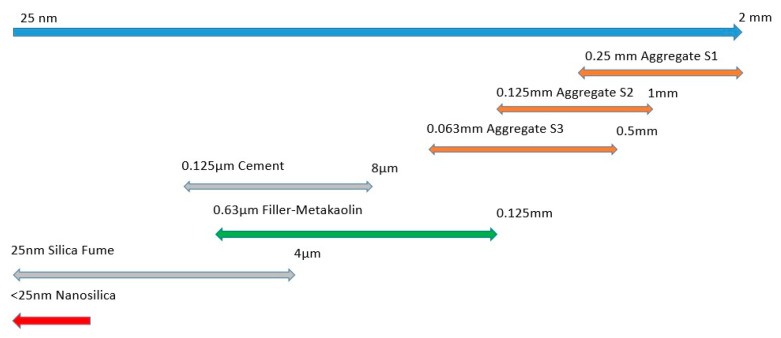
Size of aggregates and additions used.

**Figure 11 nanomaterials-11-01414-f011:**
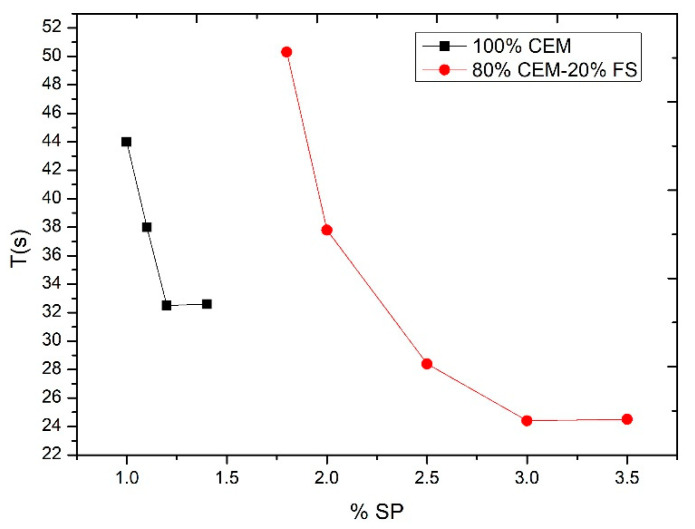
Saturation point of additive.

**Figure 12 nanomaterials-11-01414-f012:**
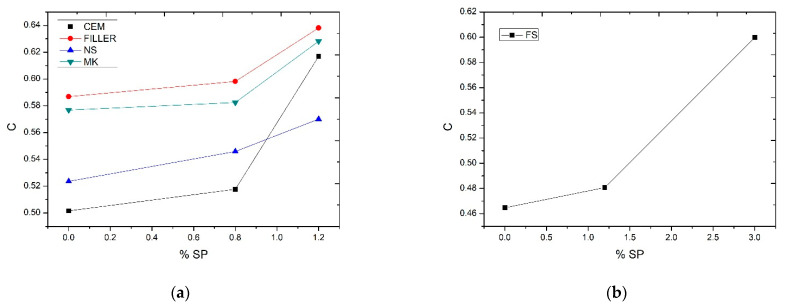
Experimental compactness of additions according to % of additive. Cement, Filler, Nanosilice, Metakaolin vs Superplasticizer (**a**); Silica Fume vs Superplasticizer (**b**).

**Figure 13 nanomaterials-11-01414-f013:**
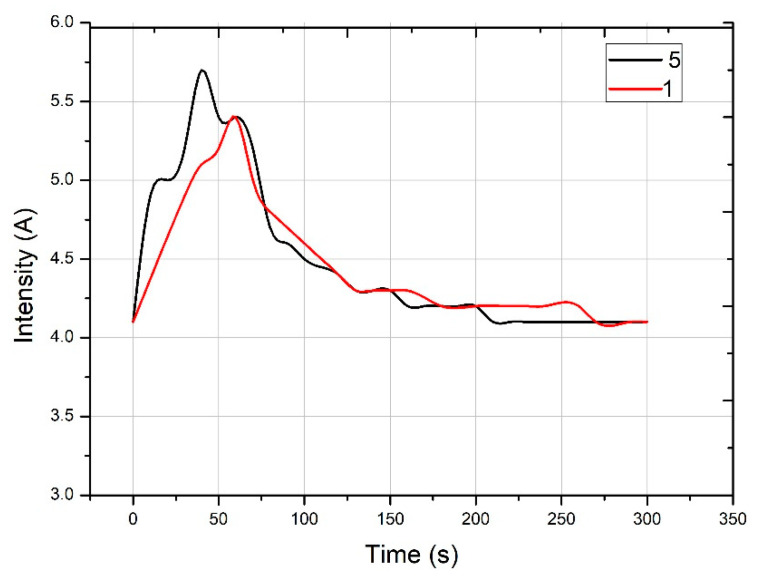
Intensity of electricity vs. time of mixing.

**Figure 14 nanomaterials-11-01414-f014:**
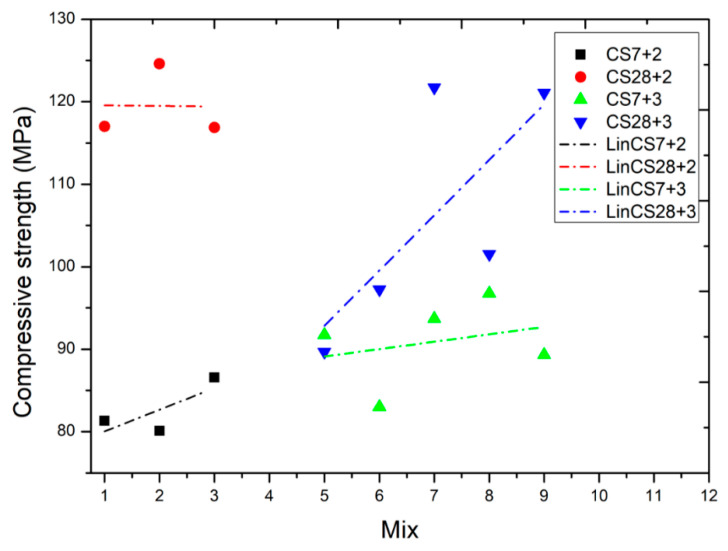
Compressive strength results.

**Figure 15 nanomaterials-11-01414-f015:**
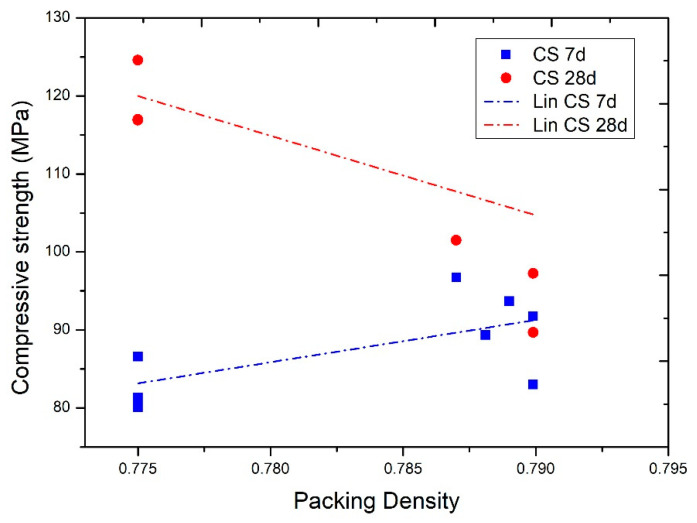
Compressive strength results according to compactness.

**Figure 16 nanomaterials-11-01414-f016:**
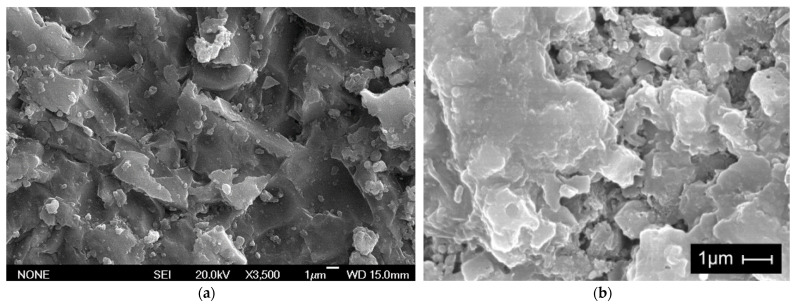
(**a**) SEM image of the C-S-H gel of concrete of mix #9 (metakaolin + nanosilica + silica fume) at the age of 43 days. (**b**) SEM image of the C-S-H gel of concrete made with CEM I 52.5 R at the age of 28 days.

**Figure 17 nanomaterials-11-01414-f017:**
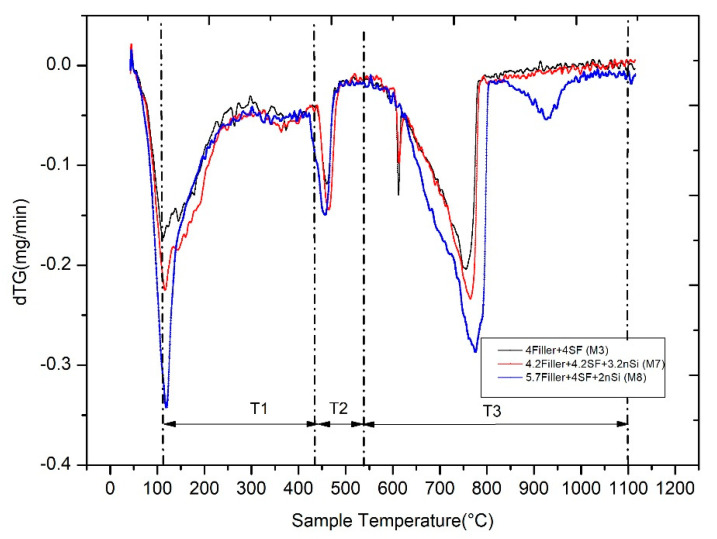
DTG curves of Mixtures 3, 7 and 8 at the age of 28 days.

**Table 1 nanomaterials-11-01414-t001:** Horsfield model of packing.

Sphere (n°)	Relative Diameter	Packing (%)
1	1	74.0
2	0.414	79.3
3	0.225	81.0
4	0.177	84.2
5	0.116	85.1

**Table 2 nanomaterials-11-01414-t002:** Compaction rates by type of packing.

Type of Packing	Packing Mode	*K*
Dry packing	Simple stacking	4.1
	Manual compaction with bar	4.5
	Vibration	4.75
	Vibration + pressure (10 KPa)	9
Wet packing	Water demand	6.7
Virtual packing		∝

**Table 3 nanomaterials-11-01414-t003:** Compaction achieved according to model compressible packing.

Mix	Aggreg 1 (S_3_)	Aggreg 3 (S_1_)	Cement	Filler Limestone	Metakaolin	Sílica F.	nSi	*φ*
1	34.1	22.7	35.2	4		4	0	0.7750
2	34.1	22.7	35.2	4		4	0	0.7750
3	34.1	22.7	35.2	4		4	0	0.7750
5	27.1	18	43.3	4.1		4.2	3.3	0.7899
6	27.1	18	43.3	4.1		4.2	3.3	0.7899
7	27.1	18	43.3	4.2		4.2	3.2	0.7890
8	27.1	17.9	43.3	5.7		4	2	0.7870
9	27.1	19.6	43.3		4	4	2	0.7881

**Table 4 nanomaterials-11-01414-t004:** Components of each mixture, compaction index and compactness.

Mix	1	2	3	5	6	7	8	9
φ	0.775	0.775	0.775	0.7899	0.7899	0.789	0.787	0.7881
K	6.45	6.45	6.45	6.45	6.45	6.45	6.45	6.45
Aggreg 1 S_3_	27.2	27.2	27.2	20.7	20.7	20.7	20.6	21.5
Aggreg 3 S_1_	18.1	18.1	18.1	13.8	13.8	13.8	13.8	14.3
Cement	28	28	28	33.2	33.2	33.2	33.2	33.2
Fíller	3.2	3.2	3.2	3.2	3.2	3.2	4.5	
Sílica F	3.2	3.2	3.2	3.2	3.2	3.3	3.1	3.1
Metakaolín								3.1
nSi				2.5	2.5	2.4	1.5	1.5
Air	2	2	2	2	2	2	2	2
Water	16	16	16	19	19	19	19	19
Additive SP	2.3	2.3	2.3	2.3	2.3	2.3	2.3	2.3
Mixing time (s)	270	270	270	300	300	300	540	480
R7 cub (MPa)	81.3	80.1	86.6	91.7	83	93.7	96.8	89.3
R28 cub (MPa)	117	124.6	116.9	89.7	97.3	121.7	101.5	121.1

**Table 5 nanomaterials-11-01414-t005:** TG results of Mixtures 3, 7 and 8 at the age of 28 days.

	Mix 3	Mix 7	Mix 8
Mass 45 °C (g)	65.0048	67.5777	67.4689
Water of equivalent Portlandite	1.03607	0.93726	1.63098
Water of total Portlandite	1.54907	1.34826	2.20598
% Water of CSH gel (140)	0.032982	0.026503	0.02851684
% Water of free Portlandite	0.007892	0.006082	0.00852245
% Water of total Portlandite	0.02383009	0.01995126	0.03269625
% Water of carbonates	0.03887405	0.03382773	0.0589605
CSH Gel (140)/Free Portlandite	4.17933723	4.35766423	3.34608696
CSH gel (140)/Total Portlandite	1.38405624	1.3283788	0.87217473
% Water of CSH gel (140)/cement content	0.00117793	0.00079828	0.00085894
% Water of free Portlandite/cement content	0.00028185	0.00018319	0.0002567

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
