# Peer review of "Achieving Ultra-High Performance Concrete by Using Packing Models in Combination with Nanoadditives"

_nanomaterials, 2021, doi:10.3390/nano11061414_

Round 1

Reviewer 1 Report

The authors studied about the compacting effect using nanomaterials in UHPC.

However, I was very confused to understand of this work.

First, when I saw the title of this paper, I thought this paper was related to numerical analysis such as FEM. I read chapter 1 and 2, this study seems to establish a new model or combine the other models. But In chapter 3, Did the authors explain the story about experiment? It was very confusing.

If this study was related to numerical analysis, then the authors had to clearly indicate the numerical analysis method and the programs that the authors used.

If this study was related to experimental things, then the authors had to clearly indicate the experimental method and the process of experiment

The expression of this study cannot understand. Really confusing.

If this study wanted to tell both the experimental and numerical analysis, the authors had to indicate the all method what I said before.

Second, the authors used very many times the word "Expression". Why chose this word? Usually, The word "Equation" or "Eq" is used for indicating the equations.

Page 7, line 204. Typo that phi_i/phi_i* (20) : this is not correct.

eg. phi_i/phi_i of Eq (20) or phi_i/phi_i of equation (20)

Page 11, line 285, "andhigher" : spacing error

Page 15, Table 3, the results of phi : 

Did these results were derived by programs? or calculated manually?

Page 15, lines 374-399 :

If this study was related to experimental results, the authors had to include the author's own experiment results such as XRF, XRD and SEM-EDS for supporting the derived results.

If this study was related to numerical results, these lines no need to indicate. 

Author Response

Comments and Suggestions for Authors

The authors studied about the compacting effect using nanomaterials in UHPC.

However, I was very confused to understand of this work.

First, when I saw the title of this paper, I thought this paper was related to numerical analysis such as FEM. I read chapter 1 and 2, this study seems to establish a new model or combine the other models. But In chapter 3, Did the authors explain the story about experiment? It was very confusing.

An adapted numerical analytical method has been used, using the equations developed by De Larrard, the central equation being (18) as indicated in the text. These equations seek the highest compactness of aggregates, cement and additions. However, the paper aims to indicate that greater compactness is not a guarantee of greater mechanical properties, and the compactness sought must be balanced with the influence on the hydration of the nanoadditions.

If this study was related to numerical analysis, then the authors had to clearly indicate the numerical analysis method and the programs that the authors used.

No specific calculation program was used to obtain the compactness. The required data on granulometry, dry and wet compactness of aggregates and additions were obtained by means of laboratory tests. Subsequently, these data were implemented in the equations mentioned in the previous point and the compactness of the mixture was calculated using an calculation sheet

Section 4 of the manuscript has been divided into two sections in order to clarify the laboratory process and results.

If this study was related to experimental things, then the authors had to clearly indicate the experimental method and the process of experiment

As it said above, Section 4 of the manuscript has been divided into two sections in order to clarify the laboratory process and results.

The expression of this study cannot understand. Really confusing.

A better explanation of the methods and procedures used has been included in the Introduction. As indicated in the first point, it is the adaptation of existing models on compactness (physical aspect) to take into account  the effectiveness of nanoadditions from the physical (compactness) and chemical (contribution to C-S-H gels) point of view. The conclusion was that the maximum compactness does not always lead to maximum strength.

The title of Section 2 has been modified in order to clarify its content:  “Theoretical background: Packaging models”

If this study wanted to tell both the experimental and numerical analysis, the authors had to indicate the all method what I said before.

The extension of the wording of the Introduction intends to include the information required by the reviewer

Second, the authors used very many times the word "Expression". Why chose this word? Usually, The word "Equation" or "Eq" is used for indicating the equations.

 The word "expression" has been replaced by "equation".

Page 7, line 204. Typo that phi_i/phi_i* (20) : this is not correct.

  1. phi_i/phi_iof Eq (20) or phi_i/phi_iof equation (20)

The equation is correct, it establishes the relationship between the actual compactness (volume of the solid)  of a particle "i" and the maximum actual compactness (volume of the solid) in the presence of other particles  *. If particle "i" is dominant, both coincide, giving an infinite or zero compaction index value according to equation (21), indicated in bibliographical reference 26.

Page 15, Table 3, the results of phi : 

Did these results were derived by programs? or calculated manually?

The data are calculated with a calculation sheet as explained in previous paragraphs.

Page 15, lines 374-399 :

If this study was related to experimental results, the authors had to include the author's own experiment results such as XRF, XRD and SEM-EDS for supporting the derived results.

In the article, the above-mentioned experiments have been left out of this scope.  The authors rely on the results of other research reported in the bibliography. Although some SEM images of some of the samples have been included in order to show the effect of the nanoadditions in the microstructure of the hydrated products.

If this study was related to numerical results, these lines no need to indicate. 

The authors really appreciate the reviewer comments that help to improve the quality of the manuscript.

Reviewer 2 Report

The manuscript presents the packing models used for the design of UHPC. The use of a combination of nanoadditions was proposed for this task. I found the article is well written; methodology and results are well described and graphically presented. Therefore this is an interesting paper in the journal scope and could be published. Although I have an overall good feeling with this paper, there are some points which need revision:

(1) The scientific discussion is very limited, usually describing the numbers of the tests, but without looking or providing for an explanation on the differences. Therefore, the article's discussion of the results should be developed.

(2) Figure 10 lacks fly ash as one of the main active pozzolanic additive used in concrete.

(3) Does the proposed models take into account reactive aggregates or applies only to inert aggregates ? Please elaborate this issue.

(4) Regarding section 3. Please provide SEM images or diagrams showing the processes of C-S-H phase formation as a result of the action of reactive nanoadditives.

(5) Please provide the full names of the ordinate axes in Figures 13 and 14.

(6) In the introduction section, the authors describe the advantages of nanomaterials used in composites. However, this topic has already been the subject of publication in the Nanomaterials journal and not only. It is therefore required that the authors comment on the results of previous new papers. In the introduction section, the following article should be discussed and cited:

Energies 2020, 13, 2184.

Author Response

Comments and Suggestions for Authors

The manuscript presents the packing models used for the design of UHPC. The use of a combination of nanoadditions was proposed for this task. I found the article is well written; methodology and results are well described and graphically presented. Therefore this is an interesting paper in the journal scope and could be published. Although I have an overall good feeling with this paper, there are some points which need revision:

(1) The scientific discussion is very limited, usually describing the numbers of the tests, but without looking or providing for an explanation on the differences. Therefore, the article's discussion of the results should be developed.

The discussion of the results has been extended for a better understanding of the results and conclusions

 (2) Figure 10 lacks fly ash as one of the main active pozzolanic additive used in concrete.

This addition has not been used in the present research as it has not been included in Figure 10.

 (3) Does the proposed models take into account reactive aggregates or applies only to inert aggregates ? Please elaborate this issue.

In the compactness calculation, both active and inert materials have been taken into account.

(4) Regarding section 3. Please provide SEM images or diagrams showing the processes of C-S-H phase formation as a result of the action of reactive nanoadditives.

An SEM image of one of the mixtures (Figure 15)   has been included. The figure shows the morphology of the hydrated cement in combination with metakaolin, nanosilica and silica fume (mixture #9) . In addition, the SEM imagen of the CEMI without nano addictions has also been included (Figure 16) for comparison.

Aquí probablemente podamos incluir una imagen SEM de un  cemento CEM I 52,5 R sin adiciones. Espero poder enviarla hoy.

 (5) Please provide the full names of the ordinate axes in Figures 13 and 14.

The changes have been introduced in the new version of the paper.

 (6) In the introduction section, the authors describe the advantages of nanomaterials used in composites. However, this topic has already been the subject of publication in the Nanomaterials journal and not only. It is therefore required that the authors comment on the results of previous new papers. In the introduction section, the following article should be discussed and cited:

Energies 2020, 13, 2184.

The above suggestion is included both in the text and in the bibliography under review.

The authors really appreciate the reviewer comments that help to improve the quality of the manuscript.

Reviewer 3 Report

In this paper, the packaging models that are fundamental for the design of ultra-9 high-performance concrete (UHPC), and their evolution were described.  In addition, a relationship between virtual and real compactness is obtained, through the compaction index, which may simulate the energy of compaction that the particles undergo in their placement in the mold. The cement hydration process affected by nanoadditions and the ensuing effectiveness in the properties in both fresh and hardened state according to the respective percentages in the mixture should also be studied. Generally, the topic is interesting, and the research is professionally carried out and the manuscript is well written. This paper is therefore recommended for publication. However, the following issues should be addressed:

1 The important research results should be included in the abstract section.

2 The description about effect of hydration of nanoadditions is insufficient. Maybe the book (Nano-Engineered Cementitious Composites: Principles and Practices. Springer. 2019) and paper (Nano-core effect in nano-engineered cementitous composites. Composites Part A. 2017, 95; Effect and mechanisms of nanomaterials on interface between aggregates and cement mortars. Construction and Building Materials. 2020, 240;Effect investigation of nanofillers on C-S-H gel structure with Si NMR spectra. Journal of Materials in Civil Engineering. 2019, 31(1);Investigating pore structure of nano-engineered concrete with low-field nuclear magnetic resonance. Journal of Materials Science. 2021, 56) can work as reference.

3 The authors mentioned “Initially, three aggregates with sizes S3 (0.5-1.6 mm), S2 (0-1 mm), S1 (0-0.5 mm) were used in the design of the granular skeleton.”. Does the size of aggregates determine according to the packaging models?

4 The mixing proportion by mass and detailed preparation of the developed UHPC in this paper should be provided in detailed.

5 According to the results in Table 6, the developed UHPC in this paper is not true UHPC because the compressive strength does not meet the requirements of the specification.

6 The literatures are a little old. References 32, 36,37, 40, 43,46 should be replaced with recent literatures.

7 There are several typos and vague sentences in the manuscript, so the manuscript should be carefully checked and thoroughly modified.

Author Response

Comments and Suggestions for Authors

In this paper, the packaging models that are fundamental for the design of ultra-9 high-performance concrete (UHPC), and their evolution were described.  In addition, a relationship between virtual and real compactness is obtained, through the compaction index, which may simulate the energy of compaction that the particles undergo in their placement in the mold. The cement hydration process affected by nanoadditions and the ensuing effectiveness in the properties in both fresh and hardened state according to the respective percentages in the mixture should also be studied. Generally, the topic is interesting, and the research is professionally carried out and the manuscript is well written. This paper is therefore recommended for publication. However, the following issues should be addressed:

1 The important research results should be included in the abstract section.

The abstract has been reviewed for including thehe above suggestion

2 The description about effect of hydration of nanoadditions is insufficient. Maybe the book (Nano-Engineered Cementitious Composites: Principles and Practices. Springer. 2019) and paper (Nano-core effect in nano-engineered cementitous composites. Composites Part A. 2017, 95; Effect and mechanisms of nanomaterials on interface between aggregates and cement mortars. Construction and Building Materials. 2020, 240;Effect investigation of nanofillers on C-S-H gel structure with Si NMR spectra. Journal of Materials in Civil Engineering. 2019, 31(1);Investigating pore structure of nano-engineered concrete with low-field nuclear magnetic resonance. Journal of Materials Science. 2021, 56) can work as reference.

The authors want to thank the reviewer for the proposed references. Such references have been included in the introduction section.

3 The authors mentioned “Initially, three aggregates with sizes S3 (0.5-1.6 mm), S2 (0-1 mm), S1 (0-0.5 mm) were used in the design of the granular skeleton.”. Does the size of aggregates determine according to the packaging models?

UHPC concretes are designed, in order to obtain a higher compactness, with low maximum sizes of aggregates, since a larger particle size leads to a lower compactness in the mixture, as indicated in the bibliographical reference 2 (A. Naaman and K. Will). Therefore, these aggregate sizes have been used based on the UHPC bibliographical references and the experience of the authors. Nevertheless,  UHPC have been obtained with sizes up to 8 mm, as mentioned in the same bibliographical reference, but most recent references uses smaller aggregate size. 

4 The mixing proportion by mass and detailed preparation of the developed UHPC in this paper should be provided in detailed.

The above suggestion has been included in the text.

5 According to the results in Table 6, the developed UHPC in this paper is not true UHPC because the compressive strength does not meet the requirements of the specification.

The results correspond to concrete without the addition of fibres. In subsequent tests of concrete with the same granular skeleton but with a fibre addition of 2% by volume, compressive strength results of more than 150 MPa were obtained. Specifically, among others, with the addition of limestone filler, silica fume and nanosilica. These clarifications are included in the revised manuscript. In addition,  the PCA (Portland Cement Association) reference where it specifies the strength range of UHPC concrete between 120-150 MPa has been included.

6 The literatures are a little old. References 32, 36,37, 40, 43,46 should be replaced with recent literatures.

New references have been included. Since the other references may be useful for the reader, they have not been removed. Nevertheless, the authors would have no objection in removing such references.

7 There are several typos and vague sentences in the manuscript, so the manuscript should be carefully checked and thoroughly modified.

The manuscript has been reviewed by a native reviewer with expertise on technical papers, especially focused on concrete technology. In addition, during this review process some additional typos have been found.

The authors really appreciate the reviewer comments that help to improve the quality of the manuscript

Reviewer 4 Report

General Comment 1

The manuscript presents a literature review on packaging models to design ultra-high-performance concrete (UHPC). In addition, an experimental study is presented where nine concretes mix designs are studied based on a discrete packaging model, considering the effect of fillers, nanoparticles and also the cement hydration process. Some properties of the obtained concretes, for both the fresh and hard states, were obtained and compared, such as the compaction index, compactness, mixing-time versus electricity intensity, and compressive strength.

The topic of the manuscript is very interesting since the concrete mix design method highly affects the final mechanical performance of the material. In addition, more optimized mix design methods can allow for a reduction of the cement consumption, which can contribute to mitigate the carbon emissions due to cement production. The results of the presented study could constitute a base for optimized concrete mix design methods, namely for UHPC.

I made some comments in order to improve the manuscript. The authors should take the comments into account and revise their manuscript.

General Comment 2

The manuscript is focused on UHPC. However, the obtained and reported concrete compressive strengths in Table 6, with a minimum value of 89.7 MPa and a maximum value of 124.6 MPa at 28 days, does not reach the minimum value (150 MPa) defined by the well-known and recognized ACI Committee 239 and also reference [1] to be classified as UHPC. In the introduction section, the authors state that “Other definitions refer to compressive strength of at least 120 MPa”, but no references justify this statement.

In order to avoid confusion for the readers, the title and content of the article should be reviewed in order to not particulate the described and used packaging models to design specifically UHPC.

Specific Comment 1

Abstract

I consider that the abstract does not reflect well the content of the article and should be revised. The abstract should clearly summarize the objectives, the methodology, the main results and conclusions. For instance, no clear reference to the experimental study was made.

Specific Comment 2

Introduction

The introduction section is short and somewhat poor. This section must clearly describe the objectives of the research, refer previous related studies with some discussion of the results, relate the presented research with such previous studies, explain what is the novelty, need and real contribution of the presented article. Please revise and improve the introduction section.

Specific Comment 3

References

The references must be cited in the text by numerical order (please refer to the guide for authors).

Specific Comment 4

Introduction

A reference related with the “ACI Committee 239” is missing. Also, as referred in the General Comment 2, statement “Other definitions refer to compressive strength of at least 120 MPa” should be justified or simply deleted to avoid confusion.

Specific Comment 5

As for the figures, all tables must be cited in the text before they appear. Units are missing in some tables. Also, substitute “Tabla” by “Table” in the caption of Table 2. Please revise.

Specific Comment 6

In some figures the text is very hard to read, for instance in Figure 1. Please check all figures and correct this issue.

Specific Comment 7

Both the title of Section 4 and the structure of this section are not adequate. Since it refers to a new part of the article, including an experimental study from the authors, it must be revised. Please, see similar studies and their structures.

For instance, Section 4 should incorporate an introductory paragraph and several specific subsections describing the used methodology, mix design details, used materials, experimental procedures, results and discussions, …

Specific Comment 8

Conclusions

The extensive paragraph before the conclusions section should be condensed and inserted in the conclusions section, as an introductory paragraph before the numbered conclusions.

Author Response

Comments and Suggestions for Authors

General Comment 1

The manuscript presents a literature review on packaging models to design ultra-high-performance concrete (UHPC). In addition, an experimental study is presented where nine concretes mix designs are studied based on a discrete packaging model, considering the effect of fillers, nanoparticles and also the cement hydration process. Some properties of the obtained concretes, for both the fresh and hard states, were obtained and compared, such as the compaction index, compactness, mixing-time versus electricity intensity, and compressive strength.

The topic of the manuscript is very interesting since the concrete mix design method highly affects the final mechanical performance of the material. In addition, more optimized mix design methods can allow for a reduction of the cement consumption, which can contribute to mitigate the carbon emissions due to cement production. The results of the presented study could constitute a base for optimized concrete mix design methods, namely for UHPC.

I made some comments in order to improve the manuscript. The authors should take the comments into account and revise their manuscript.

General Comment 2

The manuscript is focused on UHPC. However, the obtained and reported concrete compressive strengths in Table 6, with a minimum value of 89.7 MPa and a maximum value of 124.6 MPa at 28 days, does not reach the minimum value (150 MPa) defined by the well-known and recognized ACI Committee 239 and also reference [1] to be classified as UHPC. In the introduction section, the authors state that “Other definitions refer to compressive strength of at least 120 MPa”, but no references justify this statement.

The results correspond to concrete without the addition of fibres. In subsequent tests of concrete with the same granular skeleton but with a fibre addition of 2% by volume, compressive strength results of more than 150 MPa were obtained. Specifically, among others, with the addition of limestone filler, silica fume and nanosilica. These clarifications are included in the revised manuscript. In addition, the PCA (Portland Cement Association) reference where it specifies the strength range of UHPC concrete between 120-150 MPa has been included.

In order to avoid confusion for the readers, the title and content of the article should be reviewed in order to not particulate the described and used packaging models to design specifically UHPC.

A better explanation of the method and procedure used has been included in the Introduction. It is the adaptation of existing models on compactness (physical aspect) to take into account  the effectiveness of nanoadditions from the physical (compactness) and chemical (contribution to C-S-H gels) point of view. The conclusion is that maximum compactness does not always lead to maximum strength.

Specific Comment 1

Abstract

I consider that the abstract does not reflect well the content of the article and should be revised. The abstract should clearly summarize the objectives, the methodology, the main results and conclusions. For instance, no clear reference to the experimental study was made.

The summary is modified taking into account the above proposal.

Specific Comment 2

Introduction

The introduction section is short and somewhat poor. This section must clearly describe the objectives of the research, refer previous related studies with some discussion of the results, relate the presented research with such previous studies, explain what is the novelty, need and real contribution of the presented article. Please revise and improve the introduction section.

The introduction is amended to considering the above suggestion.

Specific Comment 3

References

The references must be cited in the text by numerical order (please refer to the guide for authors).

The references are amended taking into account the above suggestion.

Specific Comment 4

Introduction

A reference related with the “ACI Committee 239” is missing. Also, as referred in the General Comment 2, statement “Other definitions refer to compressive strength of at least 120 MPa” should be justified or simply deleted to avoid confusion.

The reference is included, the explanation concerning the strength of concrete is included in the first answer.

Specific Comment 5

As for the figures, all tables must be cited in the text before they appear. Units are missing in some tables. Also, substitute “Tabla” by “Table” in the caption of Table 2. Please revise.

The indication is modified in the revised manuscript

Specific Comment 6

In some figures the text is very hard to read, for instance in Figure 1. Please check all figures and correct this issue.

The indication is modified in the revised manuscript

.

Specific Comment 7

Both the title of Section 4 and the structure of this section are not adequate. Since it refers to a new part of the article, including an experimental study from the authors, it must be revised. Please, see similar studies and their structures.

For instance, Section 4 should incorporate an introductory paragraph and several specific subsections describing the used methodology, mix design details, used materials, experimental procedures, results and discussions.

Section 4 has been divided into two sections in order to clarify the laboratory process and results 

Specific Comment 8

Conclusions

The extensive paragraph before the conclusions section should be condensed and inserted in the conclusions section, as an introductory paragraph before the numbered conclusions.

A paragraph is included in the conclusions summarising the last paragraph before the conclusions.

With reference to the copyright in figure 4:

 * Figure 4 is an adaptation of the figure in the following quote from the bibliography. This thesis is freely accessible at the following address.

https://pastel.archives-ouvertes.fr/tel-01289611/file/TH2016PESC1001_complete.pdf

The authors really appreciate the reviewer comments that help to improve the quality of the manuscript

Round 2

Reviewer 1 Report

The authors did revise this paper and improve.

But I felt The supporting data is not enough to explain the author's results.

First, phi_i/phi_iof Eq (20) or phi_i/phi_iof equation (20)

The equation is correct, it establishes the relationship between the actual compactness (volume of the solid)  of a particle "i" and the maximum actual compactness (volume of the solid) in the presence of other particles  *. If particle "i" is dominant, both coincide, giving an infinite or zero compaction index value according to equation (21), indicated in bibliographical reference 26.

  • I did not point that the equation was wrong. I suggested change your "expression style" like "phi_i/phi_i of Eq (20)" or "phi_i/phi_iof equation (20)".

Second, The authors added SEM images for explaining and supporting the results, but it is not enough. Figure 13 and 14, the authors indicated just trend. Of course, SEM image showed well the C-S-H gels and shape of that. But The authors did not check the changing (e.g. 3, 7, 14, 28 days of curing) of the cement matrix from chemical analysis. So I said XRD, XRF, SEM-EDS.

If the authors checked the variation of chemical content or characteristics, the authors could explain why the compactness did not lead the compressive strength changing.
Also, the authors wanted to say that the experiment data applies to introduced model, but if the authors confirmed the chemical characteristics of this experiment, then the authors were able to deliver the contents way better than now in paramteric aspect.

Sorry about rejection, but I don't feel this paper has enough quality to publish in this journal.

Reviewer 2 Report

I have no comments.

Author Response

The authors really appreciate the work of the reviewer 

Reviewer 4 Report

I´m satisfied with the authors’ replies to my comments and I also consider that most of my suggestions and concerns have been properly considered by the authors to improve the article.

Author Response

Tha authors really appreciate the work of the reviewer.